# Semi-Synthetic Dihydrotestosterone Derivatives Modulate Inherent Multidrug Resistance and Sensitize Colon Cancer Cells to Chemotherapy

**DOI:** 10.3390/pharmaceutics15020584

**Published:** 2023-02-09

**Authors:** Ferenc István Nagy, Dóra Izabella Adamecz, Ádám Baji, Ágnes Kiricsi, Ildikó Huliák, Andrea Rónavári, Zoltán Kónya, Éva Frank, Mohana Krishna Gopisetty, Mónika Kiricsi

**Affiliations:** 1Department of Biochemistry and Molecular Biology, Doctoral School of Biology, University of Szeged, Közép Fasor 52, H-6726 Szeged, Hungary; 2Department of Organic Chemistry, University of Szeged, Dóm tér 8, H-6720 Szeged, Hungary; 3Department of Oto-Rhino-Laryngology, Head and Neck Surgery, University of Szeged, Tisza Lajos krt. 111, H-6720 Szeged, Hungary; 4Department of Applied and Environmental Chemistry, University of Szeged, Rerrich tér 1, H-6720 Szeged, Hungary; 5Molecular Genetics, German Cancer Research Center (DKFZ), Im Neuenheimer Feld 280, 69120 Heidelberg, Germany

**Keywords:** multidrug resistance, drug-resistant cancer, ABC transporters, combination therapy, pyrimidine-fused androgens, endoplasmic reticulum stress

## Abstract

Multidrug resistance (MDR) is a serious hurdle to successful cancer therapy. Here, we examined the efficiency of novel semi-synthetic dihydrotestosterone derivatives, more specifically androstano-arylpyrimidines in inhibiting the efflux activity of ATP-binding cassette (ABC) transporters and sensitizing inherently MDR colon cancer cells to various chemotherapy drugs. Using the Rhodamine123 accumulation assay, we evaluated the efflux activity of cancer cells following treatments with androstano-arylpyrimidines. We found that acetylated compounds were capable of attenuating the membrane efflux of inherently MDR cells; however, deacetylated counterparts were ineffective. To delineate the possible molecular mechanisms underlying these unique activities of androstano-arylpyrimidines, the degree of apoptosis induction was assessed by AnnexinV-based assays, both upon the individual as well as by steroid and chemotherapy agent combination treatments. Five dihydrotestosterone derivatives applied in combination with Doxorubicin or Epirubicin triggered massive apoptosis in MDR cells, and these combinations were more efficient than chemotherapy drugs together with Verapamil. Furthermore, our results revealed that androstano-arylpyrimidines induced significant endoplasmic reticulum stress (ER stress) but did not notably modulate ABC transporter expression. Therefore, ER stress triggered by acetylated androstano-arylpyrimidines is probably involved in the mechanism of efflux pump inhibition and drug sensitization which can be targeted in future drug developments to defeat inherently multidrug-resistant cancer.

## 1. Introduction

An estimated 12–14 million new cancer cases are diagnosed every year and ca. 10 million deaths per year are attributed to some kind of cancer worldwide [1]. Although several treatment approaches, such as surgery, chemotherapy, radiation therapy, immunotherapy, and targeted therapy, are available, many of them are ineffective against multidrug-resistant (MDR) cancer [2]. If the tumor manifests resistance against several structurally and mechanistically different drugs, it is considered multidrug-resistant. Therefore, MDR usually represents a major obstacle to effective therapeutic interventions against cancer [3,4,5]. According to one classification, there are two types of MDR: intrinsic and acquired drug resistance. In this concept, drug resistance can occur due to the activation of pre-existing/intrinsic/inherent mechanisms or as a result of acquired mechanisms [5]. In the case of acquired resistance, typically repeated drug administrations generate the selection pressure that leads to a reduction of anticancer agent potency, where the appearance of the antineoplastic agent is responsible for the activation of certain evolutionary mechanisms that ultimately increase the divergence of transformed cancerous cells within the tumor tissue [5,6,7]. The selection pressure does not have to be introduced exogenously, as in the case of chemotherapy, it can be inherently present in the body prior to any treatment [5,6,8,9]. Intrinsic/inherent resistance can be initiated by inherited genetic alterations that result in cancer cells manifesting reduced responses to chemotherapy and target drugs independent of any prior exposure to the therapeutic agent [10]. Regardless of the origin of the selection pressure, the result is often the emergence of a cancer cell population that is irresponsive to the effects of one or many chemotherapeutic agents [11].

Several molecular mechanisms contribute to the MDR phenotype, such as genetic mutations, epigenetic alterations, reduced drug uptake, drug sequestration or neutralization, altered drug metabolism, evasion of apoptosis, increased efficiency of DNA repair, and the activation of specific signaling pathways; nevertheless, the overexpression of certain efflux pumps has been shown to be largely responsible for the intensified drug export across the plasma membrane of cancer cells [12]. In this respect, the superfamily of ATP-binding cassette (ABC) transporters stands out. These transporters are physiologically present, as seen in the liver, gastrointestinal tract, blood-brain barrier, and kidneys, and their role is to translocate a variety of endobiotic and xenobiotic molecules through biological membranes [13]. Cancerous cells might also express a number of these membrane pumps (e.g., P-glycoprotein (Pgp, MDR1, ABCB1), multidrug-resistance protein 1 (MRP1, ABCC1), and breast cancer resistance protein (BCRP, ABCG2)) since many chemotherapy drugs are substrates of the above-mentioned transporters and, thereby, drugs can be expelled from cancer cells. Therefore, the overexpression and/or functional enhancement of ABC transporters is often associated with poor clinical outcomes [14]. Clearly, inhibiting the function of certain ABC transporters would result in an extended timeframe and opportunity for the antineoplastic drugs to exert their intracellular anticancer activity, which could significantly improve patients’ response to therapy [15]. For this reason, three generations of ABC transporter inhibitors (called chemosensitizers or MDR modulators) have been developed and tested, but unfortunately Phase III clinical trials of these compounds yielded poor results [16].

Recently, several steroidal modulators (both natural and synthetic ones) of MDR that were capable of attenuating the efflux activity of membrane transporters efficiently [17] were designed and introduced to cancer therapy. For example, progesterone and its derivatives, as well as glycocholic acid and modified deoxycholic acid derivatives, were found to inhibit ABCB1 [18,19,20], budesonide and mifepristone modulated MDR through interaction with ABCC1 [21,22], and beclomethasone along with 6a-methylprednisolone inhibited ABCG2 [23]. Moreover, several novel hybrid compounds carrying fused sterane and pyrimidine ring systems, synthesized and tested by ourselves, exhibited strong anticancer activity on a number of different cancerous cells, induced endoplasmic reticulum stress, manifested an impressive potential to inhibit ABCB1, and consequently sensitized ABCB1-overexpressing MCF-7/KCR human breast adenocarcinoma cells to doxorubicin-induced killing [24,25].

The endoplasmic reticulum (ER) is a major site of cellular protein homeostasis. Disturbances of any kind in the protein homeostasis mechanisms lead to accumulated misfolded proteins inside the endoplasmic reticulum, and thus, the cell suffers from endoplasmic reticulum stress (ER stress). To cope with ER stress, the cell initiates the so-called “unfolded protein response” (UPR), an evolutionarily conserved mechanism that aims to restore normal ER functions. Initially, the UPR is a pro-survival mechanism, however severe or prolonged UPR induces apoptosis in cells. There are three main branches of the ER stress response. These are initiated by ER transmembrane proteins (activating transcription factor 6—ATF6, inositol-requiring kinase 1—IRE1, PKR-like eukaryotic initiation factor 2α kinase -PERK) that can sense the number of misfolded proteins in the ER and start a signal transduction pathway. Under normal ER circumstances, these proteins are bound to the binding immunoglobulin protein (BIP, also known as glucose-regulated protein 78—GRP78), which is a heat-shock protein. When misfolded proteins accumulate, BIP dissociates from the transmembrane sensors to facilitate correct folding inside the ER, activating these sensors at the same time. When activated, ATF6 is transported to the Golgi apparatus and cleaved, after which it can act as a transcription factor. IRE1α cleaves an mRNA named XBP1t to XBP1s, which encodes a transcription factor X-box-binding protein 1 (XBP1). PERK indirectly activates activating transcription factor 4 (ATF4), yet another transcription factor. All of the above-mentioned transcription factors upregulate genes, such as glucose-regulated protein 94 (GRP94), an abundant chaperon in the ER that aid the cell to restore normal ER functions. It has been reported that NF-κB is also activated during UPR, which promotes survival and drug resistance. To lower the number of misfolded proteins in the ER, endoplasmic reticulum-associated degradation (ERAD) is initiated, where misfolded proteins are transported back to the cytosol and degraded by the proteasome. This process is accelerated by the ER degradation enhancing α-mannosidase-like protein (EDEM). If ER stress is severe, the UPR takes a pro-apoptotic turn. For example, ATF4 activates the CCAAT/enhancer-binding protein homologous protein (CHOP), which promotes apoptosis [26,27,28,29].

In our previous projects, we presented numerous pieces of evidence to validate the possible attenuation of drug resistance in MDR breast cancer cells subjected to ER stress [25,30]. ER stress could be induced by treating ABCB1-overexpressing MCF-7/KCR MDR breast adenocarcinoma cells with silver nanoparticles of 75 nm diameter or with different dihydrotestosterone derivatives. MCF-7/KCR cells were developed from MCF-7 cancer cells by increasing doxorubicin selection pressure, and therefore, they could be considered an in vitro model for acquired MDR [31]. The great performance of semi-synthetic dihydrotestosterone derivatives to sensitize drug-resistant MCF-7/KCR cells raised several questions. We wanted to find out whether the sensitization effect is limited to ABCB1-overexpressing cancer cells with acquired MDR (maybe only to MCF-7/KCR breast cancer cells) or whether it is a general feature that is exhibited on other MDR cell lines with a different anatomical origin, or a different ABC transporter profile (maybe cells with inherent MDR) as well. This question prompted us to test the action of androstano-arylpyrimidines on colon-derived cancer cells, namely Colo 205 and Colo 320 cell lines. Colo 320 cells have been shown to exhibit drug resistance, overexpress several ABC transporters, carry a mutation in the tumor suppressor protein APC, and the proto-oncogene c-myc is amplified in these cells. It is a model system for inherent MDR [10,32,33,34,35,36], while drug-sensitive Colo 205 cells were used as a control [34,35,37,38,39,40]. Thus, the primary aim of this present work was to examine the sensitizing potential of semi-synthetic androstane derivatives on inherently multidrug-resistant Colo 320 colon adenocarcinoma cells. For this, we utilized, in combination with these novel steroids, five different antineoplastic agents that are routinely applied upon clinical chemotherapy. Our experiments revealed that several steroid compounds were capable of diminishing the efflux activity of inherently MDR Colo 320 adenocarcinoma cells and sensitizing them to chemotherapy drugs. Since our previous results implied that the induction of ER stress might contribute to the MDR-attenuating effects of semi-synthetic steroid derivatives, apart from the expression levels of various ABC transporters, we also examined the levels of some key ER stress markers following treatments. Our results indicate that ER stress is in fact triggered in these types of multidrug-resistant colon cancer cells upon exposure to acetylated androstano-arylpyrimidines and that these novel steroids are capable of inhibiting ABC transporter activity, thereby attenuating drug resistance and consequently sensitizing MDR cancer cells to drug-induced apoptosis.

## 2. Materials and Methods

### 2.1. Steroids and Chemotherapy Agents

The Biginelli-type multicomponent access to androstano-arylpyrimidine 17-acetates (compounds labeled by number **10**) and 17-OH derivatives (compounds labeled as **11**) under microwave irradiation was published by our group in Baji et al. [24]. The original compound numbers used in Baji et al. were retained in the present manuscript for clarity and comparability.

The chemotherapy drugs Bleomycin, Carmustine, Cisplatin, Doxorubicin, and Epirubicin were obtained from the Central Pharmacy of the University of Szeged (Szeged, Hungary).

### 2.2. Cell Culture

Colo 320 and Colo 205 adenocarcinoma cell lines were obtained from the American Type Culture Collection (ATCC) and cultured in RPMI-1640 medium (Biosera, Nuaille, France) supplemented with 10% FBS, 2 mM glutamine (Biosera, Nuaille, France), and 1% penicillin-streptomycin (Biosera, Nuaille, France). Cells were maintained under standard cell culture conditions (37 °C, 95% humidity, 5% CO_2_).

### 2.3. Rhodamine 123 Accumulation Assay

Experiments were performed similarly as described first by Ludescher [41]. Colo 320 and Colo 205 cells were seeded in 6-well plates at 10^6^ cells/well density in duplicates and left to adhere overnight. The next day, the cells were treated with either compound **10a**, **10d**, **10e**, **10f**, **10g**, **11a**, **11d**, **11e**, **11f,** or **11g** at 20 μM for 24 h. Verapamil is a well-known ABC transporter inhibitor, which serves as a positive control in Rhodamine 123 accumulation tests. Therefore, in parallel experiments, cells were exposed to Verapamil at 40 μM for 2 h. After treatment, cells were washed and suspended in a serum-free RPMI-1640 medium containing 10 µM of Rhodamine 123 (RH123) (Sigma-Aldrich, St. Louis, MO, USA). After 2 h incubation, cells were washed, and the fluorescence of RH123 of at least 10,000 cells/sample was measured using the FACSCalibur™ platform. Data were analyzed and presented by FlowJo V10.0.7 software.

### 2.4. Cell Viability Assay

In order to test the viability of drug-resistant Colo 320 cells before and after treatments with chemotherapy drugs or synthetic steroids, 3-(4,5-Dimethylthiazol-2-yl)-2,5-diphenyltetrazolium bromide (MTT) cell viability assays were conducted (first described by Mosmann [42]). First the toxicity of chemotherapy drugs was determined. Colo 320 cells were seeded in 96-well plates at 10^4^ cells/well density and were left to grow for 24 h. To obtain dose-response curves, cells were treated with a serial dilution of each drug (Bleomycin, Carmustine, Cisplatin, Doxorubicin, Epirubicin; treatment was applied in 200 μL media) for another 24 h (for the applied concentrations, please refer to Appendix A). Then, the media were discarded and replaced with 100 μL fresh media containing 0.5 mg/mL MTT reagent (Sigma-Aldrich, St. Louis, MO, USA). After a 1 h incubation, this media was replaced by 100 μL DMSO to solubilize formazan crystals. Absorbance of the samples was measured at 570 and 630 nm with a Synergy HTX microplate reader (BIOTEK^®^, Santa Clara, CA, USA). Untreated cells were considered as 100% cell viability during data analysis. IC_50_ and the Hill-slope values were obtained from the measured data. The IC_10_–IC_90_ values for each drug were calculated from the IC_50_ and the Hill-slope values.

We tested also the toxicity of androstano-arylpyrimidine 17-acetates. For this, drug-sensitive Colo 205 and multidrug-resistant Colo 320 cells were exposed to compounds **10a**, **10d**, **10e**, **10f**, and **10g** in a fixed 20 μM concentration for 24 h, then MTT viability assay was performed. Cell seeding and absorbance measurement were carried out in the same way as described above.

For the combinational treatments (chemotherapy drug + steroid), each chemotherapy drug was applied in its previously determined respective IC_30_–IC_70_ concentration (in the case of each drug, the appropriate concentrations are presented in Appendix A, indicated in the first column of the table next to the drug name) together with each androstano-arylpyrimidine 17-acetate (**10a**, **10d**, **10e**, **10f**, and **10g**). The steroids were used in a fixed 20 μM concentration. Based on these treatments, certain combinations were selected to be utilized later for an apoptosis detection assay.

### 2.5. Apoptosis Detection Assay

To test the induction of apoptosis in cancer cells, Annexin V/propidium iodide assay was performed [43]. Cells were seeded at 10^6^ cells/well density in 6-well plates and left to adhere for 24 h. The next day, cells were treated with either one of the steroid compounds **10a**, **10d**, **10e**, **10f,** or **10g** at 20 μM concentration for 24 h or with one of the drugs (Bleomycin, Carmustine, Cisplatin, Doxorubicin, or Epirubicin) in individual pre-determined concentrations based on the MTT screening. According to the MTT screening, we could identify effective steroid + chemotherapy drug combinations; and from these positive hits, the ones with the lowest chemotherapy drug concentration were always regarded (see Appendix A green colored labeling). In such cases, where a given chemotherapy drug that was applied together with different steroid derivatives and the lowest efficient drug concentration to reduce the viability of cancer cells was different, then always the higher drug concentration was chosen for the apoptosis experiments. The drug concentrations for single or combination treatments upon apoptosis experiments were the following: Bleomycin 215 µM, Cisplatin 45 µM, Epirubicin 19 µM, Carmustine 603 µM, Doxorubicin 252 µM. 

To estimate the degree of membrane efflux inhibition exhibited by the synthetic steroids and how this would affect the degree of apoptosis induced in steroid+drug combination treatments, furthermore, to compare the efficiency of these steroid-induced actions with the performance of a well-known ABC transporter inhibitor (Verapamil [44]), we used Verapamil in parallel experiments. Verapamil was applied at 4 μM concentration for 24 h. 

After each treatment, media were discarded and the cells were collected and stained with Annexin V-fluorescein isothiocyanate/propidium iodide (PI), according to the manufacturer’s recommendations. The fluorescence values of 10,000 cells/sample were measured with FACSCalibur™, and the data was analyzed with FlowJo V10.0.7 software. Since the fluorescence of PI might interfere with that of Doxorubicin and Epirubicin, dot plots of Annexin V vs. forward scatter were created, rather than Annexin V vs. PI, to present the results.

### 2.6. Quantitative RT-PCR

For RNA isolation, cells were seeded in 6-well plates with 10^6^ cells/well density and left to grow for 24 h. The next day cells were treated with either **10a**, **10d**, **10e**, **10f,** or **10g** at 20 μM for 24 h. After treatments, total RNA was isolated with the RNeasy^®^ Mini Kit (QIAGEN, Hilden, Germany) according to the manufacturer’s recommendations. The concentration of the isolated total RNA was measured with a NanoDrop ND 1000 Spectrophotometer (Thermo Fisher Scientific, Waltham, MA, USA).

From each sample, 900 ng RNA was reverse transcribed in a 20 μL reaction volume using a TaqMan^®^ Reverse Transcription kit (Applied Biosystems, Thermo Fisher Scientific, Waltham, MA, USA) according to the manufacturer’s instructions.

cDNA was diluted 5× with RNase-free H_2_O to 100 μL. qPCR reactions (as in [45]) were performed on PicoReal™ Real-time PCR (Thermo Fisher Scientific, Waltham, MA, USA) using SYBR Green qPCR Master Mix (Thermo Fisher Scientific, Waltham, MA, USA). qPCR reactions were carried out in 10 μL reaction volume (5 μL SYBR Green, 3 μL RNase-free H_2_O, 1 μL cDNA, 1 μL primer-mix) in duplicates. Experiments were repeated three times. Primer sequences and final concentrations can be found in Appendix A. Relative transcript levels were determined by the ΔΔCt method [46], using GAPDH as the reference gene.

### 2.7. Immunoblotting

For immunoblotting [47], Colo 320 cells were seeded in 60 mm Petri dishes at 2.5 × 10^6^ cells/dish density and allowed to adhere overnight. The next day, cells were treated with either compound **10a**, **10d**, **10e**, **10f,** or **10g** at 20 μM for 24 h. Tunicamycin is a well-known ER stress inducer [48], therefore, it serves as a positive control in ER stress-related experiments. Colo 320 cells were exposed to Tunicamycin at 600 nM concentration for 24 h. After treatments, cell extracts were prepared using a RIPA lysis buffer (50 mM Tris (pH = 7.4), 150 mM NaCl, 1 mM EDTA, 1% Triton X-100, and 1 × Protease Inhibitor Cocktail (Sigma-Aldrich, St. Louis, MO, USA)). After centrifugation at 13,000 rpm, the supernatants were collected and their protein concentration was measured using a modified [49] Bradford method [50].

From each sample, 25 μg of total protein were separated by sodium dodecyl sulfate-polyacrylamide gel electrophoresis (SDS-PAGE) on 8 or 10% gels, then transferred to nitrocellulose membranes (Amersham^®^, Cytiva, Dassel, Germany). Before incubation of the membrane with adequate primary antibodies, the membrane was cut into two (between the two usual bands appearing at 55 and 70 kDa, respectively, of PageRuler Plus Prestained Protein Ladder (Thermo Scientific)). This way, lower amounts of diluted antibodies could be used upon incubation given the smaller-sized membranes. The part of the membrane with smaller-sized proteins (below 55 kDa) was probed with an anti-Actin antibody solution in one container. The upper part of the membrane was probed with anti-ABCB1, anti-IRE1a, and anti-BIP antibodies in another container. As for the PageRuler Plus Prestained Protein Ladder (Thermo Scientific), its bands appear on the membrane and helped us in cutting the membrane. Nevertheless, these do not appear upon scanning by the C-DiGit Blot Scanner. Membranes were blocked with 5% non-fat dry milk diluted in 0.05% TBST (20 mM Tris, 150 mM NaCl and 0.05% Tween20) for 1 h, then incubated with primary antibodies ABCB1 at 1:500 (#NB100-80870, Novus Biologicals, Centennial, CO, USA), BIP at 1:500 (#3177S, Cell Signaling Technology, Danvers, MA, USA), IRE1a at 1:1000 (#3294S, Cell Signaling Technology, Danvers, MA, USA), and beta-Actin at 1:1000 (#4970S, Cell Signaling Technology, Danvers, MA, USA) overnight at 4 °C. The next day, primary antibodies were discarded and the membranes were incubated with HRP-conjugated secondary antibodies (DAKO, Santa Clara, CA, USA) for 1 h. Membranes were probed with an ECL reagent (Millipore, Burlington, MA, USA) and visualized by the C-DiGit Blot Scanner (LI-COR, Lincoln, NE, USA). The presented images are representative blots from three individual experiments.

### 2.8. Statistical Analysis

Analysis of the acquired data was carried out using a GraphPad Prism 8.0.1 software (GraphPad Software, Inc., La Jolla, CA, USA) with Fisher’s LSD test. Differences were considered statistically significant, if *p* < 0.05 (*), *p* < 0.01 (**), *p* < 0.001 (***), *p* < 0.0001 (****); “ns” indicates non-significant.

## 3. Results

### 3.1. Androstano-Arylpyrimidine 17-Acetates Attenuate the Efflux Activity of Membrane Transporters in Colo 320 Cells

Previously, we tested acetylated (**10**) and deacetylated pairs (**11**) of androstano-arylpyrimidines and showed that androstano-arylpyrimidine 17-acetates exhibited a remarkable potential to inhibit ABCB1 activity and sensitize multidrug-resistant MCF-7/KCR breast cancer cells to Doxorubicin-induced cell death [25]. In that model system, the multidrug resistance of MCF-7/KCR breast cancer cells was acquired, as it was the result of increasing Doxorubicin concentration pressure and the consequent overexpression of ABCB1.

Therefore, in this present study, we examined the capability of acetylated and deacetylated androstano-arylpyrimidines to inhibit the efflux pump activity in another type of multidrug resistance model in the inherently MDR colon cancer Colo 320 cells. We assessed the potential of these semi-synthetic steroids to sensitize inherently MDR cells to clinically utilized chemotherapy agents. The chemical structures of the tested androstano-arylpyrimidines are shown in Figure 1. The original compound names used in our previous publication [24] detailing the synthesis of these molecules were retained here for clarity and comparability.

To achieve our goal, multidrug-resistant Colo 320 cells were treated with either of the androstano-arylpyrimidines **10a**, **10d**, **10e**, **10f**, and **10g** (acetylated compounds), or **11a**, **11d**, **11e**, **11f**, and **11g** (deacetylated compounds) at 20 μM for 24 h, then RH123 accumulation assay and cell viability screen were performed (Figure 2A,B). RH123 is a fluorescent dye and a substrate of several ABC transporters, and therefore, the intracellular concentration of RH123 is an indicator of the ABC transporter activity of a given cell type. In these latter experiments Verapamil, a known ABC transporter inhibitor [44], was used as a positive control (applied at 40 μM, for 2 h) because due to its inhibitory effect on ABC transporters, larger amounts of RH123 are retained within cells.

The RH123 accumulation experiments performed on Colo 320 cells indicated that, similarly to Verapamil, the acetylated steroidal derivatives **10a**, **10d**, **10e**, **10f,** and **10g** caused significant intracellular accumulation of RH123, indicated by higher RH123 fluorescence intensities, while their deacetylated counterparts **11a**, **11d**, **11e**, **11f,** and **11g** did not exert a similar effect (Figure 2A). Among **10a**, **10d**, **10e**, **10f,** and **10g,** the highest intracellular RH123 retention was observed upon **10e** and **10f** treatments. This means that androstano-arylpyrimidine 17-acetates are capable of inhibiting the efflux activity of MDR Colo 320 cells, but their deacetylated versions do not reduce membrane transporter activity. These results on inherently MDR colon cancer cells are in line with our previous observations on MCF-7/KCR breast adenocarcinoma cells exhibiting acquired multidrug resistance. Moreover, we could also conclude that this inhibitory feature observed for the acetylated steroids was not the result of a direct toxic effect since acetylated androstano-arylpyrimidines did not induce significant viability loss of Colo 320 cells at the applied 20 μM concentration, and only in the case of compound **10g** treatment was a slight reduction in cell viability observed (Figure 2B).

These results indicate that androstano-arylpyrimidine 17-acetates are capable of inhibiting ABC transporter activity in Colo 320 adenocarcinoma cells, but they also suggest that the efflux transporter inhibiting capacity of this compound group is manifested in breast- and colon-derived multidrug-resistant cancer cells. Importantly, it seems that the modulatory action of the acetylated androstano-arylpyrimidines on the function of ABC transporters does not depend on the type of multidrug resistance, both acquired, and inherently resistant cancer cells can be manipulated using this compound group. For further experiments, the most potent derivatives, namely **10a**, **10d**, **10e**, **10f,** and **10g,** were selected.

To compare the performance of the androstano-arylpyrimidine acetates on another cancer cell line originating from the colon tissue, like Colo 320 cells, however not manifesting multidrug resistance, we treated Colo 205 drug-sensitive colon cancer cells with **10a**, **10d**, **10e**, **10f,** and **10g** and determined the viability and the RH123 retention of these cells. We observed that apart from compound **10g**, no other acetylated androstano-arylpyrimidine exhibited toxicity on Colo 205 cells at the applied 20 μM concentration (Figure 3A). More importantly, our results revealed that Colo 205 cells exposed to **10a**, **10d**, **10e**, **10f,** and **10g** did not accumulate more RH123 than untreated control cells (Figure 3B). These findings imply that the semi-synthetic acetylated androstano-arylpyrimidine compounds have no significant effect on the membrane transporter function of drug-sensitive Colo 205 colon cancer cells. The fact that these acetylated steroid derivatives exert significant inhibition on the efflux activity only of ABC transporter-overexpressing multidrug-resistant Colo 320 cancer cells could potentially be exploited in a combination therapy, where semi-synthetic androstanes are applied together with common antineoplastic drugs.

### 3.2. Androstano-Arylpyrimidine 17-Acetates Sensitize Colo 320 Cells to Chemotherapeutic Drugs

We observed that acetylated androstano-arylpyrimidines attenuate the efflux activity of MDR cancer cells, thus these steroids, applied together with chemotherapeutic drugs, should enhance the cytotoxic performance of the chemotherapy drugs by keeping the drugs within the cancer cells, allowing the drugs to exert their molecular mechanism, and ultimately leading to an enhanced cancer cell death. In order to examine directly this drug-sensitizing effect of the acetylated androstano-arylpyrimidine compounds, we selected five clinically utilized chemotherapeutic drugs, namely Bleomycin, Carmustine, Cisplatin, Doxorubicin, and Epirubicin, and applied them on Colo 320 MDR cancer cells individually as well as in combination with the steroid compounds to determine their toxic effects.

First, Colo 320 cells were treated with each chemotherapy drug individually to determine the IC_50_ values (Table 1) using MTT cell viability assays. Based on the obtained data, the IC_10_-IC_90_ values could be calculated for each chemotherapy drug, respectively (for details please refer to Appendix A). Next, chemotherapy drug+acetylated steroid derivative combinations were tested on Colo 320 cells. In this case, the applied steroid concentration was always fixed, but the chemotherapy drug concentrations varied in a range corresponding to the IC_30_, IC_40_, IC_50_, IC_60,_ and IC_70_ concentrations of each drug. Semi-synthetic androstano-arylpyrimidine 17-acetates **10a, 10d, 10e, 10f,** or **10g** were employed at 20 μM concentration in every treatment. For example, the viability of Colo 320 cells receiving Bleomycine + **10a** combination was assessed in five different sets of measurements, as the concentration of Bleomycin was set either to 45, 68, 98, 143, or 215 μM, respectively, and at the same time cells also received compound **10a** administered every time in 20 μM concentration. This was repeated for Bleomycine + **10d,** Bleomycine + **10e,** and so on, and with the other four chemotherapy drugs (in their own IC_30_, IC_40_, IC_50_, IC_60,_ and IC_70_ concentrations), subsequently. Viability results were compared to those obtained when cells were subjected to the chemotherapy drug alone. When the drug + steroid combination resulted in a more enhanced loss of viability than the individual drug exposure, it was considered a positive hit. The pharmacological screening yielded several positive hits, the most efficient combinations at the lowest chemotherapy drug concentration are highlighted with green color in Appendix A (e.g., for Bleomycin + **10g**, with Bleomycin concentration at 45 μM). This screen formed the basis to select combinations for subsequent experiments to determine their apoptosis-inducing capacity.

Next, to verify the efficiency of the selected drug and steroid combinations and determine their apoptosis-inducing potential, an Annexin V-fluorescein isothiocyanate (FITC)-based apoptosis detection assay was performed. Colo 320 cells were treated with either a given chemotherapy drug alone, the steroid derivative (**10a**, **10d**, **10e**, **10f,** and **10g**) alone, or their appropriate combination for 24 h. In parallel experiments, the apoptosis-triggering activity of the chemotherapy drug and Verapamil (the latter applied in 4 μM for 24 h) combinations were also assessed. This way the performance and efficiency of acetylated androstanes and Verapamil can be compared, especially regarding their respective combinations with chemotherapy drugs.

Individual androstano-arylpyrimidine 17-acetate treatments resulted in no or a rather low degree of apoptotic cell death (representative dot plots are shown in Figure 4). In the case of compound **10a,** an average of only 1.89% of cells were in the Q2 quadrant and thus undergoing apoptosis (for the other compounds the percentage of apoptotic cells were the following: **10d**: 2.13%, **10e**: 5.67%, **10f**: 1.36%). Treatment with compound **10g** led to the apoptotic death of approximately 15.24% of Colo 320 cells. Exposing Colo 320 cells to individual chemotherapy drugs at the selected concentrations also did not result in massive programmed cell death since the percentage of Annexin V-positive cells is rather low in each case (Figure 4).

Although Verapamil alone induced some apoptosis in Colo 320 cells (the percentage of apoptotic cells was around 9.34%), this did not mask the apoptosis-inducing effect of Verapamil + chemotherapy drug combinations. As expected, each chemotherapy drug + Verapamil treatment resulted in a significantly higher degree of apoptotic cell death compared to exposures to the individual chemotherapeutic drug or Verapamil separately (Figure 4). The most prominent enhancement in the percentage of apoptotic cells upon chemotherapy drug + Verapamil versus chemotherapy drug treatments was observed in the case of Doxorubicin and Epirubicin.

After this, the apoptosis-inducing potential of selected acetylated androstano-arylpyrimidine compounds and chemotherapy drug combinations were tested. The selection was performed based on the results of viability screens on every steroid+chemotherapy drug combination (please see Materials and Methods and Appendix A). Figure 5 shows that compounds **10a**, **10d**, **10e,** and **10f** in combination with Epirubicin elevated significantly the percentage of apoptotic Colo 320 cells (**10a** + Epirubicin: 76.2%, **10d**+Epirubicin: 88.56%, **10e** + Epirubicin: 79.68%, **10f** + Epirubicin: 78.23%) compared to Epirubicin treatments (Figure 4). Similarly, combinational treatment of Doxorubicin with **10d** and **10g** caused a significant increase in the percentage of apoptotic cells (**10d** + Doxorubicin: 60.1%, **10g** + Doxorubicin: 49.3%) compared to the effect of Doxorubicin exposure (Figure 4). In the case of both drugs, combining them with androstano-arylpyrimidine 17-acetates resulted in a significantly stronger apoptotic cancer cell death than what the drugs induced alone (Figure 4). This finding is important given that androstane derivatives applied alone did not induce apoptosis. Interestingly, the degree of apoptosis in the above-mentioned steroid + drug combinational treatments is beyond the effect exhibited by Verapamil + drug together (Figure 4). We must note that none of the steroids induced significant apoptosis when employed jointly with Carmustine, Bleomycin, or Cisplatin. These results suggest that certain chemotherapy drugs combined with the novel semi-synthetic androstano-arylpyrimidine 17-acetates are able to sensitize multidrug-resistant Colo 320 cells to the apoptotic effects of the chemotherapy drug.

### 3.3. Androstano-Arylpyrimidine 17-Acetates Do Not Alter the Expression Levels of ABC Transporters in Colo 320 Cells

As the acetylated androstano-arylpyrimidines could sensitize multidrug-resistant cancer cells to undergo apoptosis induced by certain chemotherapy agents, we hypothesized that semi-synthetic acetylated dihydrotestosterone derivatives might modulate the efflux function of Colo 320 cancer cells by suppressing the expression of the inherently present and overexpressed ABC transporters that are generally accountable for the drug resistance in MDR cancer cells. Utilizing the CellExpress database we conducted an in silico search for every ABC transporter that has ever been identified in Colo 320 cells [51]. From this ABC transporter pool, we selected only those hits that were previously validated to export precisely those chemotherapy drugs that were used in the present study based on the UniProt database. From this list, we chose seven ABC transporters (based on CellExpress) exhibiting the highest predicted expression in Colo 320 cells. These transporters are in the order of higher to lower expression levels: ABCB1, ABCC1, ABCC4, ABCC5, ABCC10, ABCC3, and ABCG2. After this in silico approach, we performed quantitative reverse transcription polymerase chain reaction (RT-qPCR) experiments to examine whether the steroid compounds **10a**, **10d**, **10e**, **10f,** and **10g** (applied in 20 μM for 24 h) would induce any changes in the expression levels of the selected ABC transporters in Colo 320 cells. Despite the thorough database search and the data supporting the expression of ABCC3 and ABCG2 in colon cancer cells, no expression of these transporters was detected in our Colo 320 samples.

The mRNA expression levels of ABCB1, ABCC1, ABCC4, ABCC5, and ABCC10 were affected by steroid derivatives in a compound-dependent manner. Interestingly, treatments with **10a**, **10d**, **10g,** and **10f** did not lead to reduced but rather to somewhat elevated relative mRNA levels of at least one ABC transporter in each case (Figure 6A). Then we examined the expression of the ABCB1 protein, the ABC transporter with the highest basal expression level in Colo 320 cells, based on in silico data, by Western blotting. Despite the slightly increased mRNA level, we observed no significant increase in the protein level of ABCB1 in Colo 320 cells treated with either **10a**, **10d**, **10e**, **10f,** or **10g** (Figure 6B). These results indicate that although acetylated androstano-arylpyrimidines might induce a slight elevation in the transcription of some ABC transporters, they do not modulate the drug resistance features of MDR Colo 320 cancer cells by altering the protein expression of ABC transporters. Thus, the observed attenuation of drug resistance is probably not the result of modulated gene expression of membrane efflux pumps, but other molecular mechanisms must be responsible for the detected sensitization effects. In the last part of our study, we examined the potential of acetylated androstano-arylpyrimidines to trigger ER stress in MDR Colo 320 cells. Since Tunicamycin is a well-known ER stress inducer, we needed Tunicamycin to serve as a positive control in subsequent experiments. For this reason, we treated Colo 320 cells with Tunicamycin and checked whether it influenced ABCB1 protein expression or not. Figure 6B shows that Tunicamycin treatment does not modify notably the protein level of ABCB1 in Colo 320 cells.

### 3.4. Androstano-Arylpyrimidine 17-Acetates Induce Endoplasmic Reticulum Stress in Multidrug-Resistant Colo 320 Cells

The initiative to study ER stress induction by androstano-arylpyrimidine 17-acetates in Colo 320 MDR cancer cells was based on the following idea: The ER is responsible for proper protein folding and for protein targeting to the appropriate destination to exert their physiological function. These events are disturbed under ER stress. Since ABC transporters are glycoproteins and need to be targeted to the plasma membrane, therefore, ER stress is expected to disturb their folding and sorting. Previous reports from our lab revealed that treatment of MDR MCF-7/KCR breast cancer cells with compound **10g** induced ER stress since the amount of ER stress markers BIP and CHOP were significantly elevated at both mRNA and protein levels [25]. We have also revealed that the intracellular distribution of ABCB1 is disturbed due to ER stress, which can be accounted for suppressing the resistant phenotype of MCF-7/KCR cells [30]. Therefore, we hypothesized that androstano-arylpyrimdines might induce ER stress in the inherently MDR Colo 320 cells as well, which might contribute to the attenuation of drug resistance observed upon exposure to these steroid derivatives. Hence, we investigated the ER stress-inducing potential of **10a**, **10d**, **10e**, **10f,** and **10g** in Colo 320 cells. The ER stress inducer Tunicamycin [48] was used as a positive control (at 600 nM for 24 h). After treatment and sample preparation, RT-qPCR and Western blot experiments were performed to quantify the mRNA and protein levels of some key markers of ER stress, respectively. The results indicated that the mRNA expression of several ER stress marker genes (*ATF4*, *ATF6*, *BIP*, *CHOP*, *EDEM*, *GRP94*, *NF-κB*, *XBP1s*, and *XBP1t*) was increased following exposure of Colo 320 cells to the semi-synthetic androstano-arylpyrimidine 17-acetates **10a**, **10d**, **10e**, **10f,** and **10g** (Figure 7A). Furthermore, the elevated protein levels of the most prominent ER stress marker BIP further confirmed the induction of ER stress (Figure 7B). Surprisingly, all the steroid derivatives (applied in the non-toxic 20 μM concentration for 24 h) induced a higher degree of ER stress than Tunicamycin, a well-known ER stress inducer (Figure 7B).

## 4. Discussion

Cancer cells often develop multidrug resistance in order to resist the ill effects of chemotherapeutic drugs. This phenomenon frequently involves the upregulation of transmembrane drug transporters categorized under the ATP binding cassette superfamily. ABC transporters confer protection for cancer cells from a wide range of antineoplastic drugs by transporting cytotoxic compounds out of the cells and thereby keeping their cytoplasmic concentrations below effective levels. In our previously published study on ABCB1-overexpressing MCF-7/KCR breast cancer cells, we reported the ABCB1 (MDR1 or Pgp)-inhibiting properties of certain steroid derivatives, namely of androstane compounds with aryl-substituted pyrimidines fused to their A-ring. Remarkably, only acetylated androstanes possessed such an ABCB1-inhibitory potential and an impressive ER stress-inducing ability. The MCF-7/KCR cells utilized in the previous work were evolved from MCF-7 breast cancer cells under gradually increasing concentrations of Doxorubicin and were shown to overexpress mainly just ABCB1 (P-glycoprotein), and therefore, MCF-7/KCR cells represent a MDR cancer cell model, where drug resistance is acquired due to drug selection pressure. However, there are other types of MDR cancer cells, some of which exhibit inherent drug resistance, and these cells, despite being multidrug-resistant, might not manifest the same cellular and molecular features as MCF-7/KCR cells [52,53]. For example, although Pgp overexpression is a prominent, if not the most prominent, element in many MDR cancers, it is plausible that other ABC transporters might also be overexpressed under pathophysiological conditions. Thus, the multidrug-resistant character of a certain type of cancer is often the net effect of the enhanced expression and function of all these membrane efflux transporters, naturally apart from other non-transporter-related features of MDR, detailed in the introduction. Based on this, in the present study, we examined the efflux transporter-inhibitory and drug-sensitizing effects of acetylated androstano-arylpyrimidines in Colo 320 colon cancer cells, which are inherently multidrug-resistant and overexpress several types of ABC transporters.

Colo 320 cells were treated with acetylated (**10a**, **10d**, **10e**, **10f**, and **10g**) and deacetylated steroid derivatives (**11a**, **11d**, **11e**, **11f**, and **11g**) separately to verify their possible inhibitory potential on membrane drug transporters using RH123 accumulation assay. In accordance with our previous results on MCF-7/KCR cells, we observed a remarkable structure-function relationship, where only acetylated derivatives **10a**, **10d**, **10e**, **10f,** and **10g** inhibited the membrane efflux activity of colon cancer cells. Among them, **10f** showed the highest inhibition of RH123 efflux in Colo 320 cells followed by **10e**, **10g**, **10a,** and **10d**. On the other hand, as expected, no difference in RH123 accumulation was observed compared to the control when the acetylated dihydrotestosterone derivatives were applied on drug-sensitive Colo 205 cancer cells. These results verify that the enhanced RH123 retention following treatment of MDR cancer cells with the acetylated heterocyclic androstanes is in fact the consequence of inhibiting the overexpressed efflux transporters present on the plasma membrane of drug-resistant cancer cells.

It is evident that the inhibition of efflux membrane transporters in MDR cancer cells could be a big step in reversing and defeating multidrug resistance. Attenuating the activity of such membrane pumps would sensitize MDR cancer cells to the toxic effects of chemotherapeutic drugs, at least to those drugs that are recognized as substrates by one or more efflux pumps. To verify that the acetylated androstano-arylpyrimidines are capable of sensitizing MDR cancer cells to the cytotoxic actions of chemotherapy drugs, we examined the cytotoxicity and the apoptosis-triggering capacity of some chemotherapy agents (Bleomycin, Carmustine, Cisplatin, Doxorubicin and Epirubicin) alone and in combination with the semi-synthetic acetylated steroids **10a**, **10d**, **10e**, **10f,** and **10g**. The results of the toxicity screen allowed us to select the most efficient combinations of the chemotherapeutic agents (at the lowest possible concentration) and the appropriate steroid derivative. The selection was based on the most significantly enhanced cytotoxicity of the combination compared to steroid derivatives or drug monotreatments, respectively. To further confirm the chemosensitizing effect of androstano-arylpyrimidine 17-acetates and that it leads to increased apoptosis induction, we performed Annexin V-assay on Colo 320 cells. We treated these cancer cells either with the chemotherapeutic drugs at specific concentrations according to the previous cell viability assay or with **10a**, **10d**, **10e**, **10f,** and **10g**, respectively, and in parallel experiments with the appropriate combination of the chemotherapy drug and the steroid or the drug and Verapamil. As expected, the number of apoptotic cells was significantly higher when Colo 320 cells received chemotherapeutic drugs in combination with Verapamil, a known inhibitor of ABCB1, compared to individual drug treatments. More importantly, treatments with either **10a**, **10d**, **10e,** or **10f** in combination with Epirubicin as well as of **10d** or **10g** in combination with Doxorubicin have significantly raised the number of apoptotic Colo 320 cells compared to what was measured for the drug or the semi-synthetic androstane derivative individual treatments. Surprisingly, the magnitude of apoptosis induction observed upon treatments with the steroid + chemotherapy drug combinations was higher than what was caused by Verapamil + drug exposures, indicating a stronger or more efficient ABC transporter inhibition and drug sensitization capacity of our synthetic derivatives than Verapamil. Albeit we have to note that a similar level of apoptosis induction was not observed in the case of other drugs (Bleomycin, Carmustine, or Cisplatin) and steroid derivative combinations. Our in silico analysis as well as in vitro experimental data indicated that although Colo 320 cells express various ABC transporters (ABCB1, ABCC1, ABCC4, ABCC5, and ABCC10), ABCB1 is the most abundant of all. Since Doxorubicin and Epirubicin are ABCB1 substrates, it is understandable why the most remarkable enhancement of Colo 320 cell sensitivity upon semi-synthetic androstane derivative treatments occurred in the case of these drugs. These findings indicate that the sensitizing effect of acetylated androstanes in Colo 320 MDR cells could probably be attributed to an inhibitory effect exerted particularly on ABCB1 by the above-mentioned derivatives.

This astonishing drug sensitization feature of acetylated androstano-arylpyrimidines is quite encouraging and hopefully translatable to clinical settings. Nevertheless, before that step, the molecular mechanisms underlying or related to the observed attenuation of efflux activity and enhanced drug sensitization effect of these steroid derivatives have to be delineated. The first idea was to detect the gene expression levels of various ABC transporters following exposures to androstane derivatives. In fact, in one of our previous publications on Colo 320 cells treated with silver nanoparticles, we reported that the reduced expression of ABCB1 was partly responsible for the attenuated drug efflux activity by P-glycoprotein [54]. Although nanoparticles and these semi-synthetic steroids are not expected to act via the same mechanisms, the modulation of transporter expression could explain the reduced resistance of these cells. Therefore, we measured the relative mRNA levels of 7 ABC transporters ABCB1, ABCC1, ABCC4, ABCC5, ABCC10, ABCC3, and ABCG2 that are most prominently expressed in Colo 320 cells and are relevant in exporting the applied drugs out of these cancer cells, based on CellExpress and UniProt databases, respectively. The expression level of 5 of these ABC transporters, namely ABCB1, ABCC1, ABCC4, ABCC5, and ABCC10 was modulated mildly by acetylated androstano-arylpyrimidines in a compound-specific manner. ABCC3 and ABCG2 were not detectable in our samples. Contrary to what was expected, we found that treatments with steroid derivatives **10a**, **10d**, **10g,** and **10f** caused slight elevations in the relative mRNA levels of at least one ABC transporter, and only compound **10e** induced a downregulation and only in the case of ABCB1. Modest changes in the mRNA levels of a gene do not always manifest on the protein level of the gene product. Therefore, we measured the protein level of ABCB1 in Colo 320 cells treated with either **10a**, **10d**, **10e**, **10f,** or **10g** and found that the protein levels were not affected by the steroid treatments. All these findings implied that the attenuated efflux activity and the drug sensitization observed in Colo 320 cells following exposures to acetylated androstano-arylpyrimidines are probably not the results of a significantly reduced ABC transporter expression.

The endoplasmic reticulum is the major site of protein homeostasis. Perturbations in its function lead to ER stress and deregulation of the protein folding machinery. Since ABC transporters are glycoproteins, their folding and glycosylation are handled by the ER. Endoplasmic reticulum stress perturbs this machinery, which leads to a decreased targeting of ABC transporters to the plasma membrane resulting in diminished efflux activity [55]. In our previous reports, a direct connection between ER stress and inhibition of ABCB1 activity has already been demonstrated [30], therefore, we hypothesized that androstano-arylpyrimidine 17-acetates might induce endoplasmic reticulum stress in multidrug-resistant Colo 320 colon cancer cells as well, ultimately leading to a decreased efflux function. To test this hypothesis, the relative changes in the transcript levels of ER stress markers *ATF4*, *ATF6*, *BIP*, *CHOP*, *EDEM*, *GRP94*, *NF-κB*, *XBP1s*, *XBP1t*, and protein levels of the prominent ER stress marker BIP were assessed in Colo 320 cells treated either with **10a**, **10d**, **10e**, **10f,** and **10g** or with the ER stress inducer compound Tunicamycin. Our results strongly indicate that the acetylated heterocyclic androstane derivatives induce significant ER stress, supporting further the involvement of ER stress in the mechanism of attenuated membrane efflux caused by androstano-arylpyrimidine 17-acetates in Colo 320 multidrug-resistant colon cancer cells.

## 5. Conclusions

The present study proved that novel semi-synthetic acetylated A-ring-fused arylpyrimidine androstane derivatives exhibit strong ABC transporter-inhibiting potential in inherently multidrug-resistant colon adenocarcinoma cells. We verified that the efflux pump inhibitory activity of these androstane compounds is strongly dependent on their acetylation status and revealed a massive drug sensitization capacity of these compounds when applied together with routinely utilized chemotherapeutic agents. We dissected some of the possible molecular events underlying the observed cellular features and concluded that modulation of ABC transporter expression is probably not involved, but the induction of ER stress is at least partly involved in the mechanism of efflux pump inhibition and drug sensitization. As these cellular features seem to be inherently intertwined in various types of MDR cancer cells, we plan to examine the molecular and cellular details of this connection in a complex transcriptomic, proteomic, and lipidomic approach using selective inhibitors and genetically modified cells. Nevertheless, our present work provides a viable structural platform for the future design and development of a new generation of inhibitors that can defeat drug resistance and enhance the response of resistant cancer cells to clinically applied chemotherapy drugs. 

## Figures and Tables

**Figure 1 pharmaceutics-15-00584-f001:**
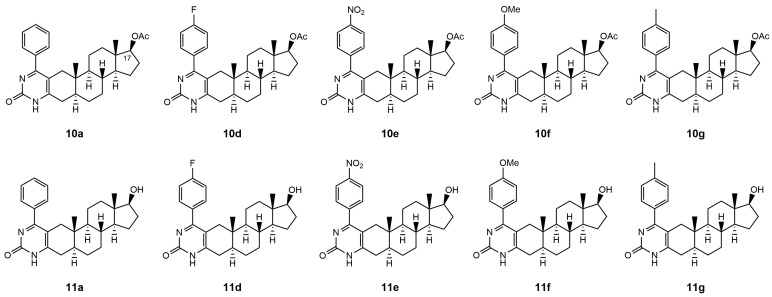
Chemical structures of androstane-derived A-ring-fused arylpyrimidines **10a**, **10d**, **10e**, **10f**, **10g**, **11a**, **11d**, **11e**, **11f,** and **11g** differing in their C-17 functionality.

**Figure 2 pharmaceutics-15-00584-f002:**
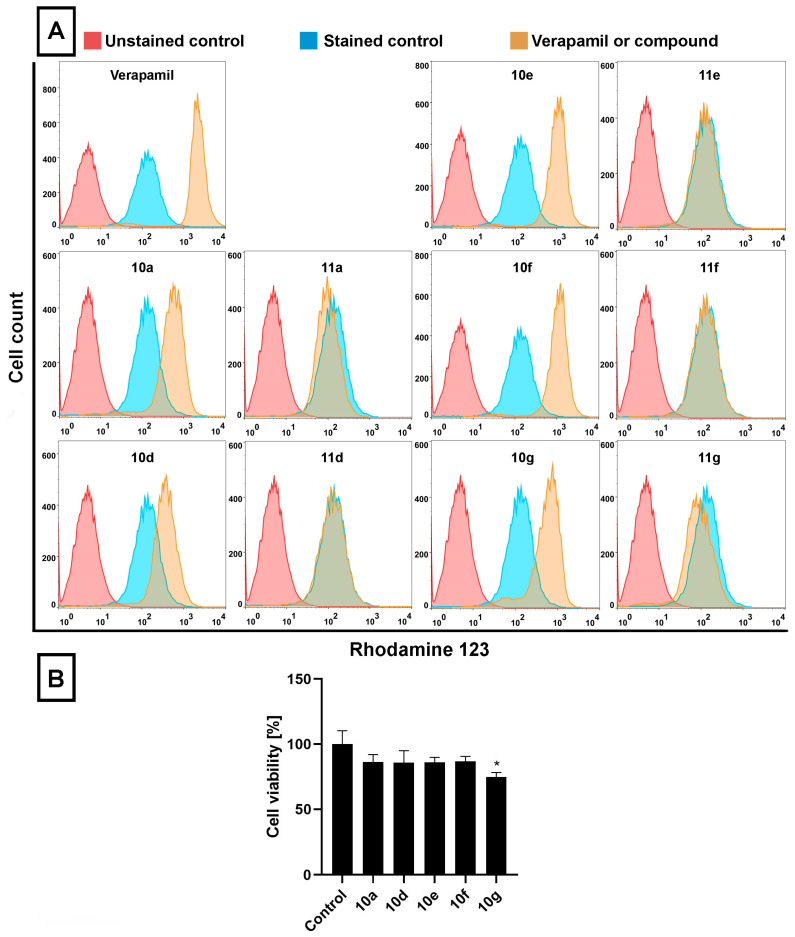
Acetylated androstano-arylpyrimidines attenuate the efflux transporter activity in multidrug-resistant Colo 320 cells. (**A**) Retention of Rhodamine 123 in Colo 320 cells treated with acetylated (**10a, 10d, 10e, 10f,** or **10g**) and deacetylated (**11a, 11d, 11e, 11f,** or **11g**) androstano-arylpyrimidines (at 20 μM concentration for 24 h) or with Verapamil (40 μM, 2 h treatment). Rhodamine 123 fluorescence of at least 10,000 cells/sample was measured by flow cytometry. (**B**) The viability of Colo 320 multidrug-resistant colon cancer cells treated with acetylated androstane compounds **10a, 10d, 10e, 10f,** or **10g** (at 20 μM, for 24 h, 10^4^ cells/well density) was assessed by MTT cell viability assay. Bar graphs represent mean ± SD values. Fisher’s LSD test, *: *p* < 0.05.

**Figure 3 pharmaceutics-15-00584-f003:**
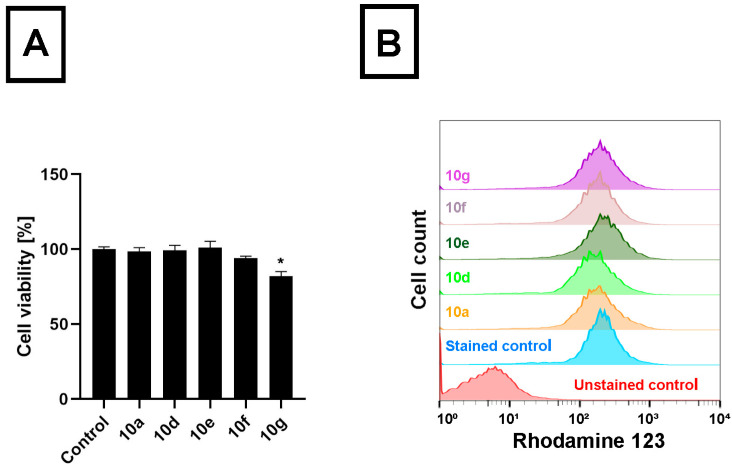
Androstano-arylpyrimidine 17-acetates do not cause Rhodamine 123 accumulation in Colo 205 cells. (**A**) Viability of Colo 205 drug-sensitive colon cancer cells following exposure to acetylated androstane compounds **10a, 10d, 10e, 10f,** or **10g** (20 μM, 24 h treatment, 10^4^ cells/well density) assessed by MTT cell viability assay. Bar graphs represent mean ± SD values. Fisher’s LSD test, *: *p* < 0.05. (**B**) Rhodamine 123 retentions of Colo 205 drug-sensitive colon cancer cells treated with acetylated androstane compounds **10a, 10d, 10e, 10f,** or **10g** (at 20 μM concentration for 24 h). Rhodamine 123 fluorescence of at least 10,000 cells/sample was measured by flow cytometry.

**Figure 4 pharmaceutics-15-00584-f004:**
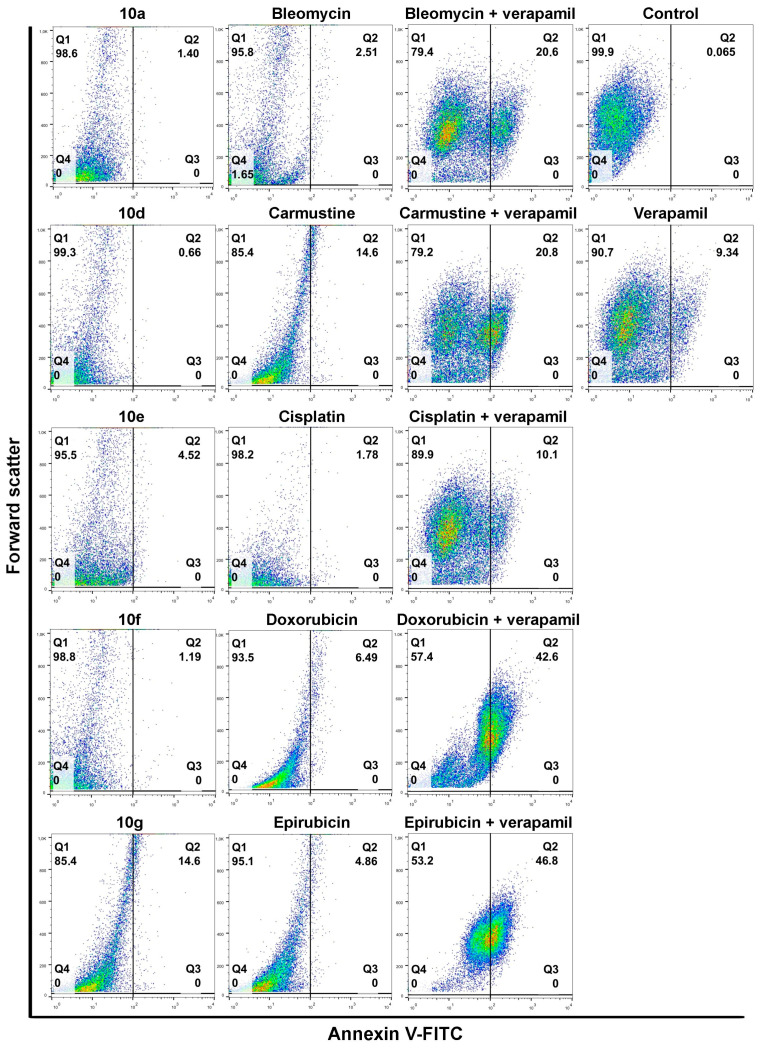
The degree of apoptosis induction in multidrug-resistant Colo 320 cells upon exposure to androstano-arylpyrimidines or various chemotherapy drugs. Annexin V-fluorescein isothiocyanate fluorescence of 10,000 cells/sample was measured with flow cytometry and plotted against forward scatter. Cells were treated either with compound **10a, 10d, 10e, 10f,** or **10g** alone (in 20 μM concentration for 24 h) or with chemotherapy drugs alone (Bleomycin, Carmustine, Cisplatin, Doxorubicin or Epirubicin at various concentrations, please refer to Materials and methods and Appendix A) for 24 h. Apoptosis induction was assessed upon Verapamil (in 4 μM for 24 h) as well as following Verapamil and chemotherapy drug combination treatments. The numbers in the Q2 quadrants represent the percentage of Annexin V-positive apoptotic cells. In the color density plots each dot represents a single detected event.

**Figure 5 pharmaceutics-15-00584-f005:**
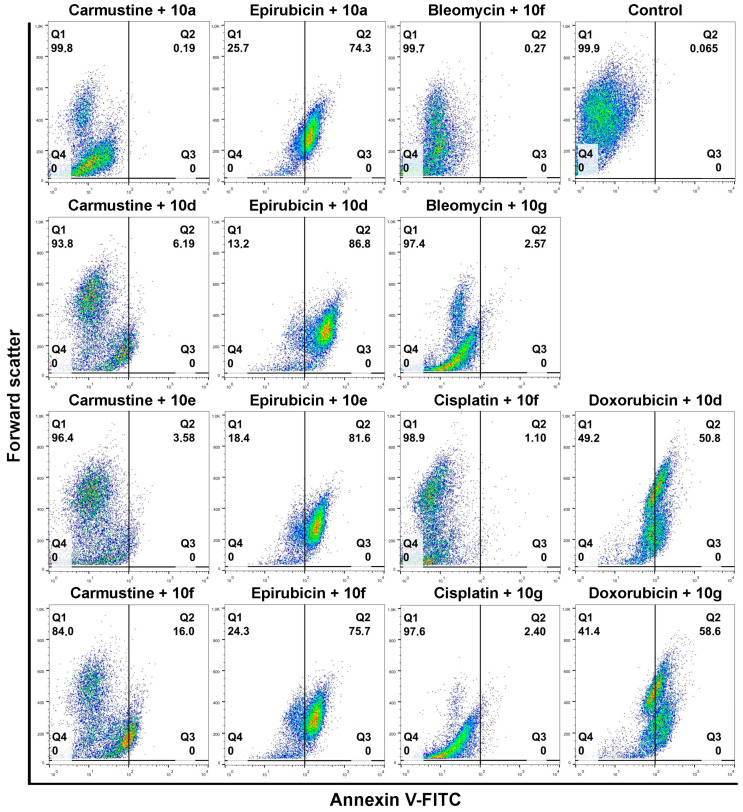
Androstano-arylpyrimidine 17-acetates sensitize Colo 320 cells to chemotherapy drug-induced apoptosis. Annexin V-fluorescein isothiocyanate fluorescence of 10,000 cells/sample were measured with flow cytometry and plotted against forward scatter. Cells were treated either with selected combinations prepared from compound **10a, 10d, 10e, 10f,** or **10g** (in 20 μM concentration for 24 h) and chemotherapy drugs (Bleomycin, Carmustine, Cisplatin, Doxorubicin, or Epirubicin at various concentrations, please refer to Materials and methods and Appendix A) for 24 h. The numbers in the Q2 quadrants represent the percentage of Annexin V-positive apoptotic cells. In the color density plots each dot represents a single detected event.

**Figure 6 pharmaceutics-15-00584-f006:**
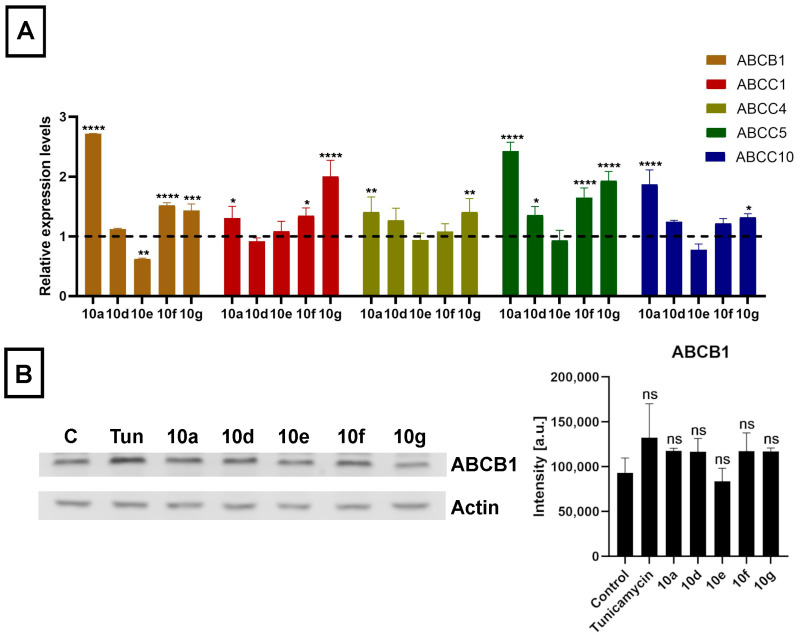
Acetylated androstano-arylpyrimidines do not suppress the efflux activity of multidrug-resistant cells by attenuating the expression of ABC transporters. (**A**) Relative mRNA levels of selected ABC transporters in Colo 320 cells treated with acetylated androstano-arylpyrimidines **10a**, **10d**, **10e**, **10f,** and **10g** (in 20 μM concentration for 24 h) determined by RT-qPCR measurements and subsequent analyses by the ΔΔCt method using GAPDH as the reference gene. For primer sequences and concentrations, please refer to Appendix A. (**B**) Representative Western blot images and densitometric analysis of ABCB1 protein levels in Colo 320 cells following exposure to androstano-arylpyrimidine acetates (treatment with steroid compounds at 20 μM for 24 h) or Tunicamycin (600 nM, 24 h). The equal loading of protein samples was verified by probing the membrane with an anti-actin antibody. C = Control, Tun = Tunicamycin. In both panels, bar graphs represent mean ± SD values. Fisher’s LSD test, *: *p* < 0.05, **: *p* < 0.01, ***: *p* < 0.001, ****: *p* < 0.0001 ns: non-significant.

**Figure 7 pharmaceutics-15-00584-f007:**
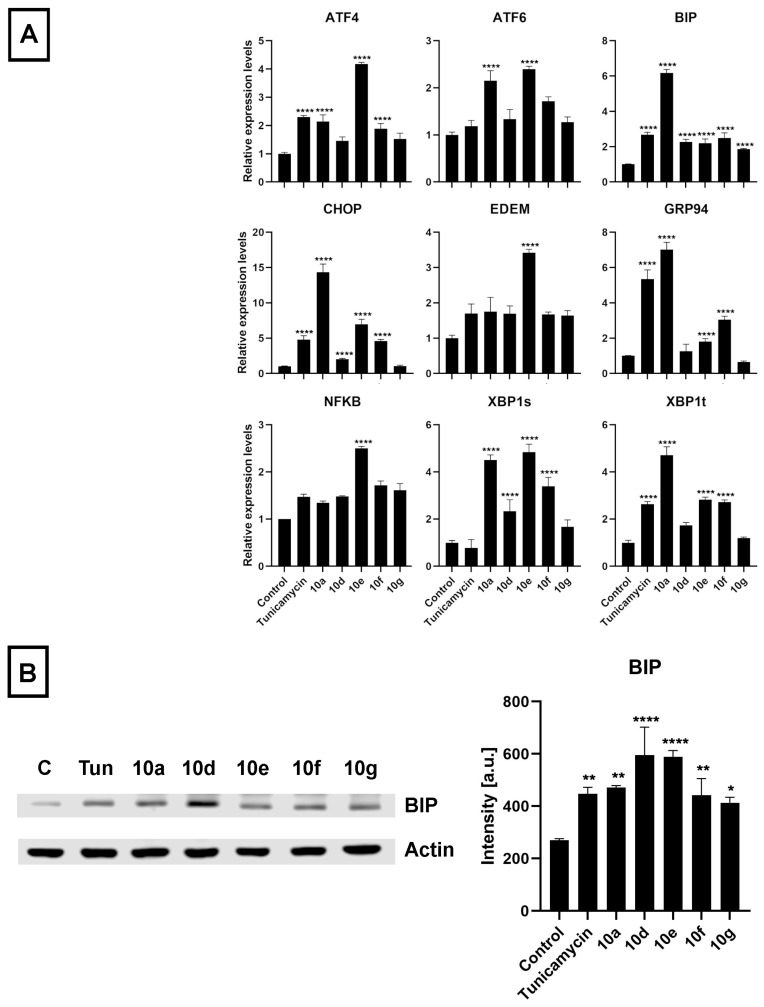
Acetylated androstano-arylpyrimidines induce endoplasmic reticulum stress in multidrug-resistant Colo 320 cells. (**A**) Bar graphs of the relative mRNA levels of various endoplasmic reticulum stress markers in Colo 320 cells treated with steroid derivatives **10a**, **10d**, **10e**, **10f,** and **10g** (in 20 μM concentration for 24 h) determined by RT-qPCR measurements and data analysis by the ΔΔCt method using GAPDH as the reference gene. For primer sequences and concentrations, please refer to Appendix A. (**B**) Representative Western blot images and densitometric analysis of the ER stress marker protein BIP in Colo 320 cells upon exposure to androstano-arylpyrimidine acetates (treatment with steroid compounds **10a**, **10d**, **10e**, **10f**, and **10g** in 20 μM, 24 h) or Tunicamycin (600 nM, 24 h). Equal loading of samples was verified by probing the membrane with an anti-actin antibody. C = Control, Tun = Tunicamycin. In both panels of figure (**A**, **B**), bar graphs represent mean ± SD values. Fisher’s LSD test, *: *p* < 0.05, **: *p* < 0.01, ****: *p* < 0.0001.

**Table 1 pharmaceutics-15-00584-t001:** IC_50_ values of selected chemotherapeutic drugs on Colo 320 cells determined by MTT cell viability assay. For the individual concentrations applied, please refer to Appendix A (cell density was 10^4^ cells/well, treatment time 24 h). Data is represented by mean ± SD.

	Bleomycin	Carmustine	Cisplatin	Doxorubicin	Epirubicin
IC_50_ [μM] ± SD	98 ± 1.07	729 ± 1.21	7.6 ± 1.63	194 ± 1.9	59 ± 1.03

## Data Availability

Data used to represent data points in case of all the graphs can be found in the Appendix A.

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
