# Peer review of "Semi-Synthetic Dihydrotestosterone Derivatives Modulate Inherent Multidrug Resistance and Sensitize Colon Cancer Cells to Chemotherapy"

_pharmaceutics, 2023, doi:10.3390/pharmaceutics15020584_

Round 1

Reviewer 1 Report

1.     This manuscript is interesting and well-done.

2.     The strength of this article is well organized for readers to understand for issue of the efficiency of novel semi-synthetic androstano-arylpyrimidines in inhibiting the efflux activity of ATP-binding cassette (ABC) transporters and sensitizing inherently MDR colon cancer cells to chemotherapy drugs.

3.     But the regretting point is wishing it was more attentive about showing STDEV is too broad in several figure. Fig. 2A (10d and 10f), Fig.6A (10f, green color), Fig.6B right (tunicamycine treated group), Fig7A (ATF4, ATF6, GRP94, XBP1t) and B (10d). If there is a large difference in the STDEV, the reliability of the data can be reduced. This point must be taken very seriously.

4.     Is there a specific reason for RNA assay instead of protein assay (ATF4/6, BIP, CHOP, EDEM GRP94, NFkB, XBP1s/1t)?

Reviewer 2 Report

In study by Nagy et al., the authors examined the efficiency of novel semi-synthetic androstano-arylpyrimidines in inhibiting the efflux activity of ABC transporters and sensitizing inherently MDR colon cancer cells to chemotherapy drugs. This study could be interesting for Pharmaceutics readers, but after significant improvements:

1.       Since ER stress markers are significant part of this study and induction of ER stress by semi-synthetic androstano-arylpyrimidines could be at least partly involved in the mechanism of efflux pump inhibition, in Introduction briefly describe involvement of ATF4, ATF6, BIP, CHOP, EDEM, GRP94, NF- κB, XBP1s and XBP in ER stress and explain connection of ER stress with drug resistance.

2.    It is not clear which concentrations of the chemotherapeutic drugs were used for single treatment in apoptosis assay. In Materials and Methods is explained that in the combined treatment chemotherapy drugs were applied at the exact same concentration as in the case of individual treatments. However, pharmacological screening, except for cisplatin and epirubicin, showed positive hits at different concentrations of chemotherapeutics. For instance, Bleomycin showed positive hits at 45µM and at 215 µM, so it is not clear which one of these concentration was used for single treatment.

3.       You should discuss more regarding combined treatment of doxorubicin and epirubicine with semi-synthetic androstane derivatives. Both chemotherapeutics are ABCB1 substrates and ABCB1 transporter has the highest basal expression level in Colo 320 cells, based on in silico data. Therefore, it is rationale to conclude that the induction of apoptosis observed only in combination of doxorubicin and epirubicin with androstane derivatives is a consequence of ABCB1 inhibition. Cisplatin, Bleomycin and Carmustin are substrates for other ABC transporter and their inhibition by androstane derivatives is not that significant.

4.      You should analyze Rhodamine 123 accumulation after short-term treatment (from 30 min till 2h) with androstane derivatives?  Based on this result you can conclude whether analyzed compounds are direct inhibitors of ABCB1 transporter or not. If androstane derivatives do not increase Rhodamine 123 accumulation after short-term treatment they are not direct ABCB1 inhibitors which could be confirmation of your conclusion that the induction of ER stress is involved in the mechanism of efflux pump inhibition.

Reviewer 3 Report

Manuscript ID: pharmaceutics-2115563

Title: Semi-synthetic dihydrotestosterone derivatives modulate inherent multidrug resistance and sensitize colon cancer cells to chemotherapy

Authors: Ferenc István Nagy, Dóra Izabella Adamecz, Ádám Baji, Ágnes  Kiricsi, Ildikó Huliák, Andrea Rónavári, Zoltán Kónya, Éva Frank,  Mohana Krishna Gopisetty *, Mónika Kiricsi *

The paper by Nagy et al. reports previously synthesized and already published semi-synthetic dihydrotestosterone derivatives. The authors show here the modulation of drug response in multi-drug resistant and sensitive colon cancer cells mediated by semi-synthetic dihydrotestosterone derivatives. They are showing that acetylated derivatives are capable to attenuate the efflux activity of multi-drug resistant cells and they are claiming that this does not result in ABC transporters modulation in expression but rather ER stress-mediated event. The authors further claim that ER stress triggered by acetylated androstano-arylpyrimidines is a fundamental molecular feature in the inhibition of ABC transporter activity which can be targeted in future drug developments to defeat inherently multidrug-resistant cancer.

Unfortunately, these forms of the manuscript and experiments performed lack mechanistic study to convince readers of statements claimed by authors. Also, it is hard to understand what is novel in this study since the authors just used another cell line type but the conclusion is the same. Therefore I cannot recommend it for publication in Pharmaceutics.

1)     The manuscript is confusing readers at the beginning. In the field of cancer treatment, MDR is defined as the ability of cancer cells to survive treatment with a variety of anticancer drugs. There are two types of MDR; one specifically carried by ATP-binding cassette (ABC) proteins and the other one which is concerned as non-ABC proteins dependent (and includes different molecular mechanisms of resistance). Here, it is not clear from the text what is already known about Colo 320 and Colo 205 regarding this issue. What does inherently in the context of MDR colon cancer Colo 320 cell line mean? What is known about these cells? Which molecular mechanisms do these cells use?

2)     Why the expression of some ABC transporters is not shown between Colo 320 and Colo 205?

3)     The figure descriptions are not informative enough and the conclusions displayed in figures’ descriptions need to be in the Results section, not in the figure description.

4)     Figure 3B needs a legend.

5)     Figure 4 and Figure 5 are not clear.  The authors need to find a more informative way of presenting those data and this presentation need to unite both figures. Figure 4 is not clear. The effect of verapamil in combination treatments is not clear since the inhibitor shows toxicity by itself. It is also not clear why the combination cisplatin/carmustine/ +verapamil was used.

6)     In Figure 6…why Tunicamycin was used? There is no explanation for this in the text.

7)     I can understand why the expression profile of different ABC transporters was measured. I do not understand, since previously was shown that these derivates do not change the expression of efflux pumps but rather their location in the membrane, why the connection between ER stress and relocation of pumps or even cell membrane intermobility was not investigated more.

8)     Also, is there any evidence that nanoparticles act in completely the same way?

9)     The experiment showing that blocking of ER stress induced by derivatives does not impact pump activity is needed (salubrinal for example).

In general, the authors are demonstrating previously published evidence about derivatives-mediated induction of ER stress and claim that their evidence support further involvement of ER stress in the mechanism of attenuated membrane efflux caused by these same compounds. The only novel thing in this story is the fact that they show that in one (only one) cell line with inherent multi-drug resistance.

I am asking why the effort was not directed at understanding the mechanisms of how ER stress is impacting efflux pumps. Which molecular mechanism is behind that regulation? This should be investigated further.

Round 2

Reviewer 1 Report

The authors have now addressed all major concerns and the manuscript is now much improved. 

Reviewer 2 Report

The authors answered to all my concerns.

Reviewer 3 Report

The revised version of the manuscript is improved.